# From Bench to Bedside in Rheumatoid Arthritis from the “2022 GISEA International Symposium”

**DOI:** 10.3390/jcm12020527

**Published:** 2023-01-09

**Authors:** Antonio Vitale, Stefano Alivernini, Roberto Caporali, Giulia Cassone, Dario Bruno, Luca Cantarini, Giuseppe Lopalco, Maurizio Rossini, Fabiola Atzeni, Ennio Giulio Favalli, Fabrizio Conti, Elisa Gremese, Florenzo Iannone, Gian Franco Ferraccioli, Giovanni Lapadula, Marco Sebastiani

**Affiliations:** 1Department of Medical Sciences, Surgery and Neurosciences, Research Center of Systemic Autoinflammatory Diseases and Behçet’s Disease Clinic, University of Siena, 53100 Siena, SI, Italy; 2Immunology Research Core Facility, Gemelli Science and Technology Park, Fondazione Policlinico Universitario A. Gemelli IRCCS, 00168 Rome, RM, Italy; 3Division of Rheumatology, Fondazione Policlinico Universitario A. Gemelli IRCCS, Università Cattolica del Sacro Cuore, 00168 Rome, RM, Italy; 4Division of Clinical Rheumatology, ASST Gaetano Pini-CTO Institute, 20122 Milano, MI, Italy; 5Department of Clinical Sciences and Community Health, Research Center for Pediatric and Adult Rheumatic Diseases (RECAP.RD), University of Milan, 20122 Milano, MI, Italy; 6Rheumatology Unit, Azienda Ospedaliera Policlinico di Modena, University of Modena and Reggio Emilia, 41121 Modena, MO, Italy; 7Rheumatology Unit, Department of Emergency Surgery and Organ Transplantations, University of Bari, 70121 Bari, BA, Italy; 8Rheumatology Unit, University of Verona, Policlinico G.B. Rossi, Piazzale A. Scuro, 37134 Verona, VR, Italy; 9Rheumatology Unit, Department of Experimental and Internal Medicine, University of Messina, 98122 Messina, ME, Italy; 10Lupus Clinic, Dipartimento di Scienze Cliniche Internistiche, Anestesiologiche e Cardiovascolari, Sapienza University of Rome, 00185 Roma, RM, Italy; 11Division of Clinical Immunology, Fondazione Policlinico Universitario A. Gemelli IRCCS, Università Cattolica del Sacro Cuore, 20123 Milano, MI, Italy; 12Scuola di Medicina, Università Cattolica del Sacro Cuore, 20123 Milano, MI, Italy

**Keywords:** inflammatory arthropathies, pathogenesis, therapy, inflammasome, synovial tissue macrophage

## Abstract

While precision medicine is still a challenge in rheumatic disease, in recent years many advances have been made regarding pathogenesis, the treatment of inflammatory arthropathies, and their interaction. New insight into the role of inflammasome and synovial tissue macrophage subsets as predictors of drug response give hope for future tailored therapeutic strategies and a personalized medicine approach in inflammatory arthropathies. Here, we discuss the main pathogenetic mechanisms and therapeutic approaches towards precision medicine in rheumatoid arthritis from the 2022 International GISEA/OEG Symposium.

## 1. Introduction

The “Gruppo Italiano di Studio sulla Early Arthritis” (Italian Group for the Study of Early Arthritis; GISEA) includes 21 hospital and community-based Rheumatology Units throughout Italy. It developed and maintains a nationwide registry to promote the study of patients with inflammatory arthritis according to standard-of-care criteria [1].

The International GISEA Meeting aims to explore the state-of-the-art in many fields of rheumatology, in particular, rheumatoid arthritis, psoriatic arthritis, and axial spondyloarthritis and joint involvement in connective tissue diseases. The present review derives from the 2022 meeting session titled “From bench to bedside in inflammatory arthropathies” and aims to summarize the main pathogenetic mechanisms and therapeutic approaches towards precision medicine in rheumatoid arthritis.

## 2. Inflammasome in the Pathogenesis of Chronic Inflammatory Arthritis

Inflammasomes consist of cytoplasmic multiprotein complexes that mediate the production of pro-inflammatory cytokines and the activation of the innate immune system. Among inflammasomes, NLRP3 is by far the most studied one; it is primarily involved in the activation of interleukin (IL)-1β and IL-18 and initiates the inflammatory cell death defined as pyroptosis. The NLRP3 inflammasome can be activated by multiple stimuli, including extracellular adenosine triphosphate (ATP), microbial toxins, reactive oxygen species (ROS), and mitochondrial DNA. All these NLRP3 agonists ultimately cause a drop in intracellular levels of potassium and chloride concentrations through the activation of ion channel gated receptors [2].

Increasing evidence shows that the NLRP3 inflammasome plays a pivotal role in recognizing innate immune signals and inducing autoreactive immune responses, probably acting as a checkpoint in innate immunity. In this regard, NLRP3 inflammasome has been found to play an important role in facilitating a variety of diseases, such as diabetes, asthma, Alzheimer’s disease, acute cerebral stroke, cancer, hypertension, myocardial ischemia, inflammatory bowel diseases, systemic lupus erythematosus, psoriasis, and different types of arthritis [3].

The most remarkable findings regarding the role of the NLRP3 inflammasome in inflammatory joint diseases were identified in the context of microcrystal arthropathies, particularly gouty arthritis; however, recent evidence demonstrates that the NLRP3 inflammasome may also play a pivotal role in the induction of rheumatoid arthritis (RA) and in the pathogenesis of seronegative spondyloarthritis (SpA).

More in detail, it is widely known how monosodium urate (MSU) stimulates abnormal IL-1 secretion by activating the NLRP3 inflammasome in patients with gouty arthritis [4]. In addition to MSU, activation of the NLRP3 inflammasome requires a second triggering factor to induce gout flares. These second factors seem to include free fatty acids and ATP flowing up from cells injured by MSU. In this context, colchicine, which is the standard of therapy for treating acute gouty arthritis, was found to inhibit the activation of the ATP-induced P2 × 7 receptor, which activates NLRP3 inflammasome by allowing potassium efflux from cytoplasm. This could explain colchicine efficacy in gout attacks [5].

Regarding RA, single nucleotide polymorphisms (SNPs) of the NLRP3 protein, a NOD-like receptor accounting for the sensor of the NLRP3 inflammasome to danger signals [6], are associated with an increased susceptibility to the disease and with a decreased response to anti-TNF agents in Caucasian patients [7]. In addition, several studies have highlighted the hyperactivity of the NLRP3 inflammasome in RA patients [8], while the inhibition of NLRP3 leads to lesser joint inflammation and bone erosion [9]. Moreover, the NLRP3 inflammasome showed to promote Th17 cell differentiation to enhance the adaptive immune dysfunction of RA [10].

Special interest has been raised by the identification of a particular pattern of SpA associated with gene mutations known to lead to inflammasome dysfunction. In particular, mutations affecting the *MEFV* gene, encoding for the pyrin, an essential regulatory component of the NLRP3 inflammasome, have been associated with a high frequency of SpA, disregarding the HLA-B27 haplotype, which may occur with a low frequency in such patients [11,12]. *MEFV* is known as the causative gene of familial Mediterranean fever (FMF), which is the first identified hereditary periodic fever and accounts for the monogenic autoinflammatory disease par excellence [13]. About 5% of FMF patients suffer from chronic articular involvement; 0.5–7.5% of adult patients may also be classified as SpA patients according to New York criteria or modified New York criteria [14]. Interestingly, FMF-SpA patients seem to have less frequent fever and more frequent arthralgia, enthesopathy, inflammatory back pain, and inflammatory bowel disease compared to patients suffering with FMF alone [15]. On this basis, SpA-FMF patients with no fever may not be as infrequent, especially in areas where FMF has a higher epidemiologic burden.

Besides patients with SpA and *MEFV* mutations, other genes may enhance the role of the NLRP3 inflammasome in SpA patients, such as NOD2/CARD15, IL1R1, and IL1R2. The number of such genes is constantly growing. Additionally, current evidence suggests the possible identification of a specific subgroup of patients with SpA mediated by inflammasome overactivation [16]. In support of this, various open label studies have pointed out that IL-1 inhibition may be somewhat effective in a non-negligible percentage of SpA patients [17,18]. Therefore, patients with high inflammatory indexes, or carrying variants known to induce inflammasome disruption, could benefit from IL-1 inhibition, especially when poorly responsive to anti-TNF or anti-IL-17 agents [17].

The role of the inflammasome as the conductor of innate immunity was also found in Behçet’s disease (BD). This disease entity is a systemic multifactorial inflammatory disorder mainly characterized by aphthous stomatitis, genital ulcers, and uveitis, with vascular, gastrointestinal, and central nervous system inflammation and perivasculitic involvement possibly occurring during patient history. It has been recently re-classified as a disorder at the crossroad of autoimmune and autoinflammatory diseases [19]. While BD was initially considered as part of the SpA continuum [18], some similarities have been disclosed in the immunopathology of these diseases [20]. While it is unclear whether BD is part of the SpA continuum, the frequency of SpA cannot be related to chance among patients with BD [20]. Anti-IL-1 inhibitors have been successfully employed in patients with BD, including those with a concomitant associated SpA [21]. This further suggests the presence of patients with SpA evolving toward an autoinflammatory phenotype that is possibly responsive to IL-1 inhibitors.

The central role of the innate immune system in the pathogenesis of Still’s disease represents a further point of reflection. Still’s disease is a protean systemic inflammatory disorder generally manifesting with high spiking fever alongside other systemic features, such as evanescent rash, arthritis, serositis, and lymphadenopathy. Still’s disease was recently classified as a multifactorial autoinflammatory disease; based on this assumption, it was successfully treated with IL-1 inhibitors. Anakinra and canakinumab have allowed for the complete and prompt control of both systemic inflammatory manifestations and of articular manifestations, including the number of tender and swollen joints and the disease activity score on 28 joints [22,23].

Based on evidence supporting the role of NLRP3 inflammasome in inflammatory joint diseases, many new possible molecules are supposed to treat inflammatory joint diseases [3,24,25,26,27,28]. In addition to anti-IL-1 agents, inhibitors of NLRP3 and caspase-1, two central proteins of the NLRP3 inflammasome, were suggested as new possible treatments for gouty and rheumatoid arthritis. Among others, tranilast, an analog of a tryptophan metabolite, was proven to inhibit the activation of the NLRP3 inflammasome in vivo and efficiently suppressed IL-1β production and neutrophil influx after tissue exposure to MSU; it also reduced acute joint swelling in murine models [24]. The β-sulfonyl nitrile OLT1177 inhibits canonical and non-canonical NLRP3 activation, as observed in monocytes from patients with cryopyrin-associated periodic syndrome and in murine models with acute arthritis [25]. Oridonin, a covalent NLRP3 inhibitor, showed preventive or therapeutic effects on mouse models of gouty arthritis [26]. Targeting caspase-1 might also represent a future treatment strategy, looking at experimental mouse models [27]. The use of IL-18 blocking antibodies could represent a further possible treatment opportunity [3,28].

## 3. Synovial Tissue Macrophage Subsets as Predictors of Drug Response in Rheumatoid Arthritis

The development of minimally invasive techniques that enable synovial tissue collection for high-throughput analysis significantly improved our understanding of RA heterogeneity in terms of synovial tissue inflammation degree and to integrate synovial tissue features in a predictive algorithm for the management of RA patients [29]. In this context, the quantification of synovial tissue inflammation degree in naive RA patients is contingent with symptom duration (being higher if the assessment is performed after 12 months from symptom onset), autoantibody positivity (being higher in ACPA and/or IgM/IgA-RF positive RA), and with treatment response to the first line treatment (being higher in RA patients who do not achieve remission after 6 months of Methotrexate therapy) [30]. Moreover, the analysis of the Pathobiology Early Arthritis Cohort (PEAC) revealed that, through histopathology, it is possible to identify different pathotypes of synovial tissue inflammation: (i) diffuse myeloid, characterized by monocyte or macrophage enrichment; (ii) lympho-myeloid, showing aggregates of B and T lymphocytes within different degrees of inflammatory cell infiltration; and (iii) pauci-immune fibroid, which is lacking a relevant inflammatory cell infiltrate [31]. In this classification, the presence of synovial tissue macrophages (STMs) is the most common feature of the joint inflammation in RA; lymphocyte enrichment and STM heterogeneity might represent a valid putative biomarker for personalized medicine in terms of treatment stratification and prognosis in RA. By using a bulk RNA-sequencing approach on paired synovial tissue and peripheral blood samples from RA patients at disease onset, different synovial tissue pathotypes are characterized by different transcripts based on the immune phenotype, suggesting that the degree of inflammation within the synovial tissue in RA is highly heterogeneous and supporting the different rate of treatment response to specific therapies [32].

It was demonstrated that some immunosuppressive treatments of RA exert their immunosuppressive action in the macrophage compartment. However, the majority of data available to date are mainly derived by in vitro studies of their effect of monocyte-derived macrophages, using peripheral blood derived cells from RA patients or by assessing of their anti-inflammatory action on mouse models of arthritis with limited evidence on humans. Among them, methotrexate was shown to increase the expression of the NF-kB suppressor A20 (TNFAIP3) in human monocyte-derived GM-CSF polarized macrophages, leading to a significant repression of pro-inflammatory cytokines [33]. Similarly, sulfasalazine was found to promote cell apoptosis and to inhibit LPS-induced TNF-α expression in human monocyte-derived macrophages [34], and leflunomide was found to inhibit pro-inflammatory cytokine release and NF-kB expression in synovial macrophages of RA patients co-cultured with Jurkat T cells [35].

Despite the early rheumatological referral and the prompt use of first-line therapeutics, nearly 40% of RA patients do not show effective response to conventional disease modifying anti-rheumatic drugs (DMARDs), making them eligible for second-line treatment [36]. Multiple biological and targeted-synthetic DMARDs represent different options for repressing synovial tissue inflammation once first-line treatment is ineffective or not tolerated. Macrophages are one of the main sources of TNF-α release in RA [37], and TNF inhibition significantly represses the migration of peripheral blood-derived Ly6C^pos^ monocytes and promotes their apoptosis within the joint of human-TNF transgenic mice [38]. A comparative analysis assessing the effect of different TNF-α inhibitors on synovial tissue macrophages in RA showed that the reduction in the number of synovial macrophages after etanercept and infliximab treatment was accompanied by an up-regulation of synovial apoptosis [39]. Interestingly, etanercept and adalimumab were found to promote the polarization of CD14^pos^ monocyte-derived M-CSF-polarized macrophages from RA patients by decreasing inflammatory surface markers (CD40, CD80) and enhancing alternative markers (CD16, CD163, and MerTK) [40]. Abatacept, a T-cell co-stimulation inhibitor, was shown to reduce the release of pro-inflammatory cytokines from monocyte-derived macrophages [41]. A seminal paper suggested a direct association between the number of synovial macrophages and joint damage extension in RA [42]. Moreover, the analysis of synovial tissue inflammatory composition in early RA revealed that scores for local disease activity are associated with the number of macrophages in the synovial sub-lining, as well as the expression of macrophage-derived cytokines as TNF-α and IL6 [43]. Various effective agents such as gold, sulfasalazine, MTX, and leflunomide can reduce the infiltration of synovial macrophages whose reduction correlates with the success of treatment response [44].

The identification of predictors of response to treatment is crucial in the context of personalized medicine, being an effective approach to optimize the rate of RA patients achieving robust and sustained remission after a given treatment. Since more than a decade ago, it was found that the immunohistochemical assessment of cell infiltration and cytokine expression in synovial tissue before the initiation of TNF-α inhibitor predict clinical response in RA patients. In particular, the synovial tissue of good responders to TNF-α inhibition RA was enriched, before treatment initiation, of macrophages such as CD163^pos^, MRP14^pos^, and MRP8^pos^ cells compared to non-responding RA [45]. Conversely, Nerviani et al. found that RA patients with a pauci-immune synovial pathotype, before treatment began, were less likely to respond to certolizumab treatment [46].

The identification of distinct tissue pathotypes of synovial tissue in the naive stage of RA dominated by different immune-cell modules have risen a key unmet need in the management of RA with the prospective identification of patients who would likely benefit from specific therapies. Recently, the results of the first biopsy-driven randomized clinical trial assessing the impact of distinct synovial tissue microanatomical organization on the rate of efficacy of distinct therapeutics targeting tissue-specific highly expressed pathways were released. Humby et al. reported that RA patients that inadequately respond to TNFi, and that are classified as having histological and RNA sequencing-defined B-cell poor synovitis, are significantly more prone to have a successful treatment response to IL6R than TNF-α inhibition [29].

The development of single-cell RNA sequencing analysis allowed for the creation of the atlas of synovial tissue macrophages in health and across different RA phases [47], showing that human synovial tissue hosts two main populations of macrophages that can be distinguished based on their expression of MerTK and mannose receptor (CD206).

In healthy human synovial tissue, resident MerTK^pos^CD206^pos^ macrophage clusters are the predominant ones, among which the TREM2^high^ cluster forms a protective lining layer, while the LYVE1^pos^ cluster mostly resides in the sub-lining layer [48]. Conversely, the synovial tissue, naive to treatment of active RA, is enriched with pro-inflammatory clusters (MerTK^neg^CD206^neg^), among which the CD48^pos^S100A12^pos^ cluster is the most abundant, expressing alarmins (i.e., S100A8, S100A9 and S100A12) and CXCL8, suggesting its ability to activate stromal cells and to promote neutrophil chemiotaxis [47].

Remission is currently the target clinical outcome in RA management [36], which is characterized by a drastic change in the phenotype of resident synovial tissue resident macrophages, whose phenotype is deeply reverted towards anti-inflammatory/regulatory MerTK^pos^CD206^pos^ STMs [47]. However, despite the achievement of sustained clinical and imaging-defined disease remission, RA patients might show histological signs of residual synovitis in terms of persistent sub-lining macrophages and T lymphocyte infiltration when compared to active RA that are naive to treatment [49], with a differential predominance of inflammation-resolving MerTK^pos^CD206^pos^ and inflammatory MerTK^neg^CD206^neg^ STMs based on the clinical definition of disease remission. In particular, the synovial tissue of RA patients in sustained Boolean remission shows higher rates of MerTK^pos^CD206^pos^ STMs than DAS-defined remission [47]. However, despite no signs of ultrasound-detected synovitis, some RA patients in sustained remission might experience disease flare up in up to 50% of cases within a year [50], suggesting the need to identify the molecular mechanisms responsible for disease flare in remission status. To date, at the synovial tissue level, the ratio between inflammation-resolving MerTK^pos^CD206^pos^ and inflammatory MerTK^neg^CD206^neg^ STMs predicts persistent remission when therapeutics are tapered and discontinued in RA in remission [47]. Since the duration of drug-free remission is very limited [51] with the progressive risk of disease flare over time [52], this clearly suggests that fragile remission is not equivalent to the self-sustained homeostasis of healthy joints. As previously described, during active RA, STMs undergo transcriptomic pathogenic changes that mostly resolve in disease remission. However, a shared set of ~30 genes, repressed in active RA, did not return to normal levels of the healthy synovium when the patients achieved remission [47]. The system biology of these ‘super-repressed’ genes indicates that they have a fundamental role for myeloid cell-mediated local immune tolerance by limiting the activation of adaptive immunity [53,54], and establishing the function of these gene pathways in the synovium might provide evidence of molecular mechanisms that lead to flare in the fragile remission state.

In conclusion, future biopsy-driven clinical studies using high-definition multiomic approaches will help to solve RA heterogeneity and will shed new light on the molecular mechanisms promoting the chronicity of synovial tissue inflammation for an individualized approach and the development of novel precision medicine algorithms for the management of RA.

## 4. The Evolution of Treatment in Rheumatoid Arthritis: The LONG Way towards a Personalized Approach

In the last 20 years, the treatment of rheumatoid arthritis (RA) has deeply changed, including the introduction of new drugs and the development of new recommendations for the management of the disease. For the first time in the history of rheumatology, remission has become a feasible outcome.

Conventional synthetic disease-modifying anti-rheumatic drugs (csDMARDs), such as methotrexate, leflunomide, and sulfasalazine, represented the standard of care for RA for many years. Over the last 20 years, the choice of possible DMARDs has expanded with the development of biologic DMARDs (bDMARDs), such as abatacept, tumor necrosis factor inhibitors (TNFi), rituximab, interleukin 6 receptor antagonists (anti-IL-6R), and, more recently, targeted synthetic (ts)DMARDs, namely baricitinib, tofacitinib, upadacitinib, and filgotinib.

The therapeutic pyramid for the treatment of RA has reversed in the last years, and guidelines for the management of RA from the European Alliance of Associations for Rheumatology (EULAR) and the American College of Rheumatology (ACR) now propose the early use of DMARDs, aspiring for remission or low disease activity [55,56]. In 2010, a treat-to-target approach was proposed for the first time [57], improving the care and outcomes of RA patients [58,59]. In the treat-to-target strategy, therapy is proposed according to therapeutic goals that are predefined in each patient [57]. However, some studies suggest that only 63% of RA patients in the USA received a DMARD. Prescribed DMARDs vary with demographic factors, socioeconomic status, and geographic location. Prescribed drugs can be influenced by low income, patients refusing treatment, comorbidities, and extra-articular manifestations of RA, resulting in contraindications to available drugs [60].

Another important point of discussion regards the possible discrepancy between real life and patients included in randomized clinical trials (RCTs). We could expect that features of patients enrolled in clinical trials may have changed in the last years, possibly influencing the results of RCTs [61]. In particular, the increased availability of treatment opportunities could induce a selection of more severe or multi-failure patients to be enrolled in RCTs or a greater inclusion of patients from low-income countries due to lower access to treatment choices.

Obviously, patient features could influence the effect of treatment, and they should be considered before translating drug safety and efficacy in clinical practice or when a post-hoc comparison between different clinical trials is performed.

Moreover, a study comparing the baseline features of RA patients enrolled in RCTs or observational studies showed better prognostic factors for people enrolled in RCTs than in observational studies, which might result in the overestimation of drug efficacy [61]. In this regard, baseline characteristics of RA patients between RCTs and observational studies showed that baseline DAS28, HAQ-DI, ESR, and CRP significantly decreased in patients participating in RCTs over the time period 1999–2015 [61].

Together with the new 2022 EULAR recommendations for the treatment of RA, the most relevant update on the treatment of RA was presented by ACR [62]. Among the novelties introduced in the new set of recommendations from ACR, MTX remains the milestone of treatment. Hydroxychloroquine or sulfasalazine are conditionally recommended only in patients with low disease activity. Biologic and ts-DMARDs are still not supported as the first strategy. Moreover, the role of glucocorticoids (GCs) is further limited; for the first time, ACR suggests their avoidance also in the initial strategy of treatment, with the only possible placement as bridging therapy when rapid symptomatic relief is needed. A dose escalation or change of DMARD is recommended when the patient is unable to withdraw systemic GCs.

EULAR recommendations differ from those published by ACR in few but significant points; first, EULAR continues to suggest the use of GCS as a bridging therapy when starting or changing DMARDS, but they should be tapered and discontinued as rapidly as clinically feasible; second, JAKi are considered first-line therapy after a csDMARD failure together with biologic DMARDs. Pertinent risk factors must be taken into account before treatment with JAKi [63]. Hydroxychloroquine may be used in patients with early, mild disease only when the other three csDMARDs are contraindicated or not tolerated. However, EULAR does not recommend that MTX be replaced by hydroxychloroquine [63].

Recently, EULAR defined “difficult-to-treat RA” and published points to consider its management [64]. In this regard, EULAR suggested an ultrasound as the main tool to confirm the inflammatory activity of disease when clinical evaluation is not sufficient; alternative diagnoses should be always considered, in particular in seronegative RA and in the early stages of disease; finally, adherence to the therapy should also be discussed and confirmed.

Moreover, the disease activity index calculated by the mean of composite measures should be considered cautiously when comorbidities occur. Fibromyalgia and obesity may affect patient-reported outcomes and the clinical assessment of the joints.

Finally, disease activity indices of RA currently have several pitfalls, including above all their complexity and their difficult application in real-world clinical settings. Therefore, the inclusion of molecular signatures reflecting the underlying mechanisms of disease activity may help to improve clinical assessment tools for RA. It might result in more accurate measurements of treatment response.

A change in the mechanism of action of DMARD should be considered after the failure of the second b/ts-DMARD, but non-pharmacological measures to manage disability, pain, and fatigue should also be proposed.

However, despite the dramatic improvement in our knowledge, precision medicine is far from being concretely applied in RA [65,66,67].

In the last 20 years, other than the increasing number of new monoclonal antibodies and small molecules, many new genetic or serological markers were discovered, improving our ability to diagnose autoimmune diseases early on.

In heterogeneous conditions, such as RA and other systemic autoimmune diseases, precision medicine could help physicians in optimizing treatment, reducing the number of switches to other therapies, and improving the adherence to therapy and overall safety of the drugs.

Until now, only few attempts have been proposed to individualize treatment in RA patients [68], but patient stratification remains quite limited. In some cases, comorbidities represent the main driver in the therapeutic choice more than the clinical and serological heterogeneity of the disease.

On the other hand, the increasing number of available monoclonal antibodies against key-cytokines in disease pathogenesis and the introduction of the new class of JAK-i made molecular-targeted treatment feasible, at least in theory.

In fact, although the treatment of RA was extensively modified in recent years, most patients fail the first therapeutic approach, many patients do not respond to methotrexate, and about 40% respond even to the first b/ts-DMARD. As above reported, about 5–20% of RA patients are resistant to all proposed treatments, and these cases are defined as “difficult to treat RA” [69].

Currently, the absence of validated and repeatable biomarkers does not allow for the early identification of non-responder people. Clinical evaluation, according to the “treat to target strategy”, remains the milestone of the therapeutic approach to RA.

Due to the high heterogeneity of RA, we can suppose that different pathogenetic pathways can be present in individual patients, and it can suggest to investigate for these different pathways to develop personalized therapies [32,70].

Although further validation is necessary, some data suggest that synovial immune signatures may be able to predict clinical responses and, in the near future, to guide treatment decisions in RA [29,31,32,46,49].

In this regard, about half of RA patients show low or absent CD20+ B cells in affected synovia. Thus, it was postulated that the level of synovial B cells/B cell-related pathways would influence treatment response to anti-CD20 monoclonal antibodies, namely rituximab. However, results from small observational studies were inconclusive and inconsistent [67].

To further explore this hypothesis, a biopsy-driven, randomized clinical trial in RA patients with inadequate response to TNFi was developed. Patients were randomized to either rituximab or tocilizumab according to synovial B cell signatures. At the end of the study, the authors reported that only 12% of patients with a low synovial B cell molecular signature had a response to rituximab, while 50% responded to tocilizumab. In contrast, in patients with high synovial B cell lineage signature, the authors did not observe differences between the two treatments. Combining histological findings and advanced molecular analyses, the authors identified genes and pathways linked to drug response. On the contrary, the lack of response to both drugs was associated with more than 1000 genes. Interestingly, the fibroid pauci-immune pathotype was associated with the poor response to the drugs, supporting the hypothesis that the pauci-immune phenotype represents a refractory endotype [71].

Although the results of these studies were nonconclusive, it demonstrated, for the first time, the possibility of developing therapeutic strategies according to individual genetic and/or histologic features.

Comorbidities and extra-articular manifestations of RA have been deeply investigated in the last ten years and largely influence the treatment of these patients. In particular, cardiovascular comorbidities and lung involvement can worsen the prognosis and the quality of life of RA patients, limiting the available therapeutic options for the rheumatologist [72,73].

In conclusion, in the next years, we expect much progress towards a precision medicine and technology, and innovations could also improve the performance of physicians in the management of autoimmune systemic diseases.

Technology will support physicians in therapeutic choice and follow-up; similarly, the development of precision medicine will allow to reduce adverse events to drugs, to increase the response to treatment, and to globally improve the retention rate of the therapy, in summary “to prescribe the best drug for the right patient early in the disease process”.

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
