# Peer review of "From Bench to Bedside in Rheumatoid Arthritis from the “2022 GISEA International Symposium”"

_jcm, 2023, doi:10.3390/jcm12020527_

Round 1

Reviewer 1 Report

This is a very interesting review reporting the involvement of the inflammasome (notably NLRP3) in inflammatory rheumatism (RA and SpA), and the role of synovial tissue macrophages in RA, including the different subsets, that may be predictive of response to treatment. Finally, the authors discuss the personalized approach of treatment in RA.

I have only a few minor comments about this review.

1-    Line 81: "to restore Treg/Th17 balance": it is not accurate. Indeed, the article cited demonstrated that JAKi restore the gama delta Treg/gama delta T17 balance which are another cells subset from Treg and Th17 cells.

2-    Line 86: "have been associated with a higher frequency of SpA".In this study the authors reported the characteristics of patients with FMF and juvenile SpA. The term SpA is therefore not correct. We would be more cautious about the association between FMF and SpA (jSpA if we refer to this article). As reported by the authors, this association may reflect a particular phenotype of FMF rather than a genuine association of jSpA and FMF.

3-    Line 108-110: "while it is unclear how BD is part of SpA continuum, the frequency of SpA cannot be related to chance among patient with BD".

In this study, the authors did not report the frequency of SpA in a population of BD patients. Indeed, the authors discussed about the sharing of a similar immunopathogenetic (association with MHC-I allele) and in some point similarity in immunopathology between these 2 diseases. However, the clinical expression of these 2 diseases is different with specific tissue involvement.

4-    Line 122: "many new possible molecules are being tested": please quote references.

5-    Line 152: "most of the therapeutics approved for the treatment of RA exert their immunosuppressive action on the macrophages compartment".

I will delete this sentence and replace it with "it has been demonstrated that some immunosuppressive treatement of RA exert their immunosuppressive action on the macrophages compartment". As reported in this review, the most studies quoted, focused on effect of immunosupressive drugs on macrophages without studied their effect on other cells as T and B cells.

6-    Line 175, 194, 218, 219, 222, 228, 232, 234, 239, 240 : please put “pos” “high” in superscript CD14pos or CD14+.

7-    Line 184: "sulfasalazine". Please correct

8-    Line 267: “anti-IL6”: it will be more accurate to talk about anti-IL-6R.

9-    Line 271: “European League Against Rheumatism “. The correct term is "European Alliance of Associations for Rheumatology". please correct

10- Line 296: 2022 EULAR recommendation for management of RA have been published recently. You can discuss about it.

11-Line 297: please cite reference of ACR guidelines (DOI: 10.1002/acr.24596)

12- Line 299: ACR guidelines recommend "conditionally" the use of hydroxychloroquine in RA in patients with low activity. I will notify the term "conditionally". Note that EULAR 2022 guidelines do not mention hydroxychloroquine because of the very low level of evidence.

13- Line 343: the cited article (tanaka, recent progress in treatments of RA) does not report these numbers "many patients don't respond to MTX and about 40% event to the first line b/Ts-DMARD. As Above reported, about 5-20% of RA patients are resistant to all proposed treatment”.Please, modify and quote the right reference.

Author Response

REVIEWER #1

1-    Line 81: "to restore Treg/Th17 balance": it is not accurate. Indeed, the article cited demonstrated that JAKi restore the gama delta Treg/gama delta T17 balance which are another cells subset from Treg and Th17 cells.

We have removed the sentence and the corresponding reference in order to avoid the misunderstanding. Thank you for the correction.

2-    Line 86: "have been associated with a higher frequency of SpA".In this study the authors reported the characteristics of patients with FMF and juvenile SpA. The term SpA is therefore not correct. We would be more cautious about the association between FMF and SpA (jSpA if we refer to this article). As reported by the authors, this association may reflect a particular phenotype of FMF rather than a genuine association of jSpA and FMF. 

We changed references. We propose, as new reference 10: Atas N, Armagan B, Bodakci E, Satis H, Sari A, Bilge NSY, Salman RB, Yardımcı GK, Babaoglu H, Guler AA, Karadeniz H, Kilic L, Ozturk MA, Goker B, Haznedaroglu S, Kalyoncu U, Kasifoglu T, Tufan A. Familial Mediterranean fever is associated with a wide spectrum of inflammatory disorders: results from a large cohort study. Rheumatol Int. 2020 Jan;40(1):41-48. doi: 10.1007/s00296-019-04412-7.

Atas et al have identified spondyloartropathy in 12.9% out of 971 patients with familial Mediterranean fever. In such a way, spondiloartropathy accounted as the most frequent comorbid inflammatory condition in subjects with familial Mediterranean fever. In these patients, the frequency of HLA-B27 aplotype was only 10.6%, suggesting that HLA-B27 does not have a major role in the pathogenesis of FMF-associated SpA. We also suggest the following reference (new reference 11):

Li Z, Akar S, Yarkan H, Lee SK, Çetin P, Can G, Kenar G, Çapa F, Pamuk ON, Pehlivan Y, Cremin K, De Guzman E, Harris J, Wheeler L, Jamshidi A, Vojdanian M, Farhadi E, Ahmadzadeh N, Yüce Z, Dalkılıç E, Solmaz D, Akın B, Dönmez S, Sarı İ, Leo PJ, Kenna TJ, Önen F, Mahmoudi M, Brown MA, Akkoc N. Genome-wide association study in Turkish and Iranian populations identify rare familial Mediterranean fever gene (MEFV) polymorphisms associated with ankylosing spondylitis. PLoS Genet. 2019 Apr 4;15(4):e1008038. doi: 10.1371/journal.pgen.1008038.

Li et al demonstrated the strong association of spondyloarthritis and the M694V mutation in the MEFV gene. This association was observed both in HLA-B27-positive and -negative cases, although it was even stronger among the latter.

We have changed the test from “Special interest has been arisen by the identification of a particular pattern of SpA associated with gene mutations known to lead to inflammasome dysfunction. In particular, mutations affecting the MEFV gene, encoding for the pyrin, an essential regulatory component of the NLRP3 inflammasome, have been associated with a higher frequency of SpA, disregarding any positive HLA-B27 haplotype [11]” into “Special interest has been raised by the identification of a particular pattern of SpA associated with gene mutations known to lead to inflammasome dysfunction. In particular, mutations affecting the MEFV gene, encoding for the pyrin, an essential regulatory component of the NLRP3 inflammasome, have been associated with a high frequency of SpA, disregarding  the HLA-B27 haplotype, which may occur with a low frequency in such patients [10,11]”.

3-    Line 108-110: "while it is unclear how BD is part of SpA continuum, the frequency of SpA cannot be related to chance among patient with BD".

In this study, the authors did not report the frequency of SpA in a population of BD patients. Indeed, the authors discussed about the sharing of a similar immunopathogenetic (association with MHC-I allele) and in some point similarity in immunopathology between these 2 diseases. However, the clinical expression of these 2 diseases is different with specific tissue involvement.

We have changed the sentence from “While it is unclear whether BD is part of the SpA continuum, the frequency of SpA cannot be related to chance among patients with BD [18]” into “While BD was initially considered as part of the SpA continuum [18], some similarities have been disclosed in the immunopathology of these diseases [19].

4-    Line 122: "many new possible molecules are being tested": please quote references.

We have added references and a brief description of molecules supposed to be useful in inflammatory joint diseases due to their direct and indirect effect on NLRP3 inflammasome. The references are included between 22 and 26.

5-    Line 152: "most of the therapeutics approved for the treatment of RA exert their immunosuppressive action on the macrophages compartment".

I will delete this sentence and replace it with "it has been demonstrated that some immunosuppressive treatment of RA exert their immunosuppressive action on the macrophages compartment". As reported in this review, the most studies quoted, focused on effect of immunosupressive drugs on macrophages without studied their effect on other cells as T and B cells.

The aim of this statement was to underline that the majority of immunosuppressive treatments used for RA management represses the activation of the macrophage compartment together with other cell types, supporting the notion that the repression of the myeloid compartment is a crucial factor for anti-inflammatory actions exerted by therapeutics. We kept the text unchanged.  

We thank the reviewer for this suggestion. We modified the text accordingly

6-    Line 175, 194, 218, 219, 222, 228, 232, 234, 239, 240 : please put “pos” “high” in superscript CD14pos or CD14+.

These were edited.

ok

7-    Line 184: "sulfasalazine". Please correct

This was corrected.

ok

8-    Line 267: “anti-IL6”: it will be more accurate to talk about anti-IL-6R.

We thank the reviewer for this suggestion. We modified the text accordingly

9-    Line 271: “European League Against Rheumatism “. The correct term is "European Alliance of Associations for Rheumatology". please correct

We thank the reviewer for this suggestion. We modified the text accordingly

10- Line 296: 2022 EULAR recommendation for management of RA have been published recently. You can discuss about it.

We thank the reviewer for this suggestion. We modified the text accordingly, adding a paragraph focused on difference between ACR and EULAR recommendations

11-Line 297: please cite reference of ACR guidelines (DOI: 10.1002/acr.24596)

We changed reference 55 accordingly

12- Line 299: ACR guidelines recommend "conditionally" the use of hydroxychloroquine in RA in patients with low activity. I will notify the term "conditionally". Note that EULAR 2022 guidelines do not mention hydroxychloroquine because of the very low level of evidence.

We thank the reviewer for this suggestion. We modified the text specifying this point

13- Line 343: the cited article (tanaka, recent progress in treatments of RA) does not report these numbers "many patients don't respond to MTX and about 40% event to the first line b/Ts-DMARD. As Above reported, about 5-20% of RA patients are resistant to all proposed treatment”.Please, modify and quote the right reference.

We thank the reviewer for showing us the mistake. We changed the reference with “Buch MH. Defining refractory rheumatoid arthritis. Ann Rheum Dis 2018;77:966–9”

Reviewer 2 Report

General aspects:

1.       Please correct the title of your article.

2.       An introduction is missing. Some information on inflammatory arthropathies in general, the specific forms of manifestation and the GISEA International Symposium would be helpful for the reader.

3.       Since the review is supposed to be about inflammatory arthropathies, it is confusing that Part 2 and und Part 3 are only about RA. You could add data about other inflammatory arthopathies in Part 2 and 3 or mention in the abstract/introduction that the focus is on RA.

4.       Please introduce all abbreviations (e.g. NLRP3, ACPA, MEFV, NOD2(CARD15).

The three parts do not seem to be fully coordinated in terms of content and language. Therefore, there will be specific comments for each part in the following:

Part 1: Inflammasome in the pathogenesis of chronic inflammatory arthritis

1.       Overall impression: This part is well written, structured and understandable.

2.       Content:

a.       A short description of Behçet’s disease and Still’s disease would be helpful for the reader.

b.      If available, more information on “the new possible molecules that are being tested in inflammatory joint diseases” and previous data on their efficacy and safety would be interesting.

3.       There are only minor language difficulties:

a.       Please make sure to use passive form in sentences like “NLRP3 inflammasome has shown” or “tofacitinib has demonstrated”.

b.      Please adjust this part: “Special interest has been arisen”, e.g. by using “raised”, “generated” or “aroused”.

c.       Please correct “per excellence” to “par excellence”.

Part 2: Synovial tissue macrophage subsets as predictors of drug response in Rheumatoid Arthritis

1.       Overall impression: This part is linguistically demanding written. However, in part it is difficult to understand as the sentences are very long and complex with multiple language errors.

2.       Content:

a.       Please clarify what Ly6Cpos cells are.

b.      Please explain what c-DMARDs are when first mentioned.

c.       What is the difference between Boolean remission and DAS-defined remission and how could these differences explain the differential predominance of certain macrophages clusters in synovial tissue?

3.       Formalities:

a.       Please highlight if “A novel taxonomy of human Synovial Tissue Macrophages (STMs) defined by single cell omics” is the headline of Figure 1.

4.       Language:

a.       Please check the use of commas. There are too many of them disrupting the reading flow and in part making the sentences not understandable.

b.      Many of the sentences are too long and complex. The sentence structure is not always clear resulting in misleading passages. Avoid using multiple participle clauses in one sentence.

c.       Please pay attention to upper and lower case letters, e.g. “Rheumatoid Arthritis”, “Methotrexate”, and “Certolizumab”.

d.      Check if singular and plural forms are used correctly, e.g. “…within different degree of inflammatory cell infiltrate…”, “The identification of distinct tissue pathotype of synovial tissue…”, “…might show histological sign of residual synovitis…”

e.      Correct “in contest” to “in context” in multiple sentences.

f.        Please make sure to use passive form in sentences like “Abatacept, a T-cell costimulation inhibitor, showed to reduce…”

g.       Please make sure to use the correct tense, e.g. “Since more than a decade ago, it was found…”

Part 3: The evolution of treatment in rheumatoid arthritis: the long way towards a personalized approach

1.       Overall impression: This section is moderately well written but make sure that sentences are not too long. This part does not seem very structured. Please structure this section better so that a common thread and key messages are apparent.

2.       Content:

a.       Please clarify what the treat-to-target approach is.

b.      It is not clear why a main part of your conclusion is about artificial intelligence and technology which you haven’t mentioned previously.

3.       There are some language difficulties:

a.       Check the use of commas. There are too many of them disrupting the reading flow.

b.      Please make sure to use the correct tense, e.g. in “…features of patients enrolled in clinical trials may be changed in the last years…”

c.       Check if singular and plural forms are used correctly, e.g. “Biologic and ts-DMARDs is still not supported…”, “…the results of these study…”

d.      The word “furtherly” does not exist.

e.      Please make sure that sentences following prepositions are in the correct form, e.g. in “although the noteworthy improvement of our knowledge”, and “despite the treatment of RA has been largely changed in the last years”.

f.        Please make sure to not forget words, e.g. “other the increasing number of new monoclonal antibodies”, “…including machine learning and artificial intelligence, could improve…”, “…reduce the treatment options, while will allow to…”, “…about 5–20% of RA patients are resistant to all proposed treatments have been defined as “difficult to treat RA”

g.       Please correct “Authors” to “authors” in multiple sentences.

h.      “to prescribe the best drug for the right patient early in the disease process” is not one word.

Author Response

REVIEWER #2

General aspects: 

  1. Please correct the title of your article. 

We thank the reviewer for this comment. We changed the tile of the article

  1. An introduction is missing. Some information on inflammatory arthropathies in general, the specific forms of manifestation and the GISEA International Symposium would be helpful for the reader.

We thank the reviewer for this comment. We added an introduction to the text

  1. Since the review is supposed to be about inflammatory arthropathies, it is confusing that Part 2 and und Part 3 are only about RA. You could add data about other inflammatory arthopathies in Part 2 and 3 or mention in the abstract/introduction that the focus is on RA. 

We thank the reviewer for this comment. We modified abstract, introduction and title specifying that rheumatoid arthritis is the main topic

  1. Please introduce all abbreviations (e.g. NLRP3, ACPA, MEFV, NOD2(CARD15).

The three parts do not seem to be fully coordinated in terms of content and language. Therefore, there will be specific comments for each part in the following: 

Part 1: Inflammasome in the pathogenesis of chronic inflammatory arthritis

  1. Overall impression: This part is well written, structured and understandable. 

Thank you for the kind appreciation of the manuscript.

  1. Content: 
  2. A short description of Behçet’s disease and Still’s disease would be helpful for the reader. 

We have inserted a brief description of both diseases.

  1. If available, more information on “the new possible molecules that are being tested in inflammatory joint diseases” and previous data on their efficacy and safety would be interesting. 

We have included some reference and more details of some molecule supposed to be a possible treatment strategy in inflammatory joint diseases. In particular, we added the references 22-26 and the following sentences: “Among others, tranilast, an analog of a tryptophan metabolite, proved to inhibit the activation of NLRP3 inflammasome in vivo and efficiently suppressed IL-1β production and neutrophil influx after the tissue exposure to MSU; it also reduced acute joint swelling in murine models [22]. The β-sulfonyl nitrile OLT1177 inhibits canonical and noncanonical NLRP3 activation, as observed in monocytes from patients with cryopyrin-associated periodic syndrome and in murine models with acute arthritis [23]. Oridonin, a covalent NLRP3 inhibitor, showed preventive or therapeutic effects on mouse models of gouty arthritis [24]”.

  1. There are only minor language difficulties: 
  2. Please make sure to use passive form in sentences like “NLRP3 inflammasome has shown” or “tofacitinib has demonstrated”. 

We have revised this aspect.

  1. Please adjust this part: “Special interest has been arisen”, e.g. by using “raised”, “generated” or “aroused”. 

We have chosen the formula: “Special interest has been raised”.

  1. Please correct “per excellence” to “par excellence”. 

 We have corrected. Thank you for your work.

Part 2: Synovial tissue macrophage subsets as predictors of drug response in Rheumatoid Arthritis

  1. Overall impression: This part is linguistically demanding written. However, in part it is difficult to understand as the sentences are very long and complex with multiple language errors. 
  2. Content: 
  3. Please clarify what Ly6Cpos cells are. 

This was specified in the text.

  1. Please explain what c-DMARDs are when first mentioned. 

This was specified as requested.

  1. What is the difference between Boolean remission and DAS-defined remission and how could these differences explain the differential predominance of certain macrophages clusters in synovial tissue?

The difference between Boolean and DAS-defined remission in RA is widely known to the Rheumatology community that relies on the level of residual inflammatory burden in terms of clinical (joint count and patient assessment) and laboratory parameters (CRP).  The reference for this statement is present in the text and we kept the sentence unchanged to limit redundancies.

  1. Formalities:
  2. Please highlight if “A novel taxonomy of human Synovial Tissue Macrophages (STMs) defined by single cell omics” is the headline of Figure 1. 

This was highlighted.

  1. Language: 
  2. Please check the use of commas. There are too many of them disrupting the reading flow and in part making the sentences not understandable. 

This was checked and edited.

  1. Many of the sentences are too long and complex. The sentence structure is not always clear resulting in misleading passages. Avoid using multiple participle clauses in one sentence. 

Sentence structure was checked.

  1. Please pay attention to upper and lower case letters, e.g. “Rheumatoid Arthritis”, “Methotrexate”, and “Certolizumab”. 

These were corrected.

  1. Check if singular and plural forms are used correctly, e.g. “…within different degree of inflammatory cell infiltrate…”, “The identification of distinct tissue pathotype of synovial tissue…”, “…might show histological sign of residual synovitis…”

       These were checked and edited.

  1. Correct “in contest” to “in context” in multiple sentences. 

These were corrected.

  1. Please make sure to use passive form in sentences like “Abatacept, a T-cell costimulation inhibitor, showed to reduce…”

This was changed.

  1. Please make sure to use the correct tense, e.g. “Since more than a decade ago, it was found…”

 This was changed.

Part 3: The evolution of treatment in rheumatoid arthritis: the long way towards a personalized approach

  1. Overall impression: This section is moderately well written but make sure that sentences are not too long. This part does not seem very structured. Please structure this section better so that a common thread and key messages are apparent. 

We thank the reviewer for this comment. We have revised the text

  1. Content: 
  2. Please clarify what the treat-to-target approach is.

We added the definition of treat-to-target strategy

  1. It is not clear why a main part of your conclusion is about artificial intelligence and technology which you haven’t mentioned previously. 

We thank the reviewer for this comment. We have revised the text

  1. There are some language difficulties: 
  2. Check the use of commas. There are too many of them disrupting the reading flow. 

We have revised the text

  1. Please make sure to use the correct tense, e.g. in “…features of patients enrolled in clinical trials may be changed in the last years…”

We modified the text

  1. Check if singular and plural forms are used correctly, e.g. “Biologic and ts-DMARDs is still not supported…”, “…the results of these study…”

We modified the text

  1. The word “furtherly” does not exist. 

We modified the text

  1. Please make sure that sentences following prepositions are in the correct form, e.g. in “although the noteworthy improvement of our knowledge”, and “despite the treatment of RA has been largely changed in the last years”. 

We modified the text

  1. Please make sure to not forget words, e.g. “other the increasing number of new monoclonal antibodies”, “…including machine learning and artificial intelligence, could improve…”, “…reduce the treatment options, while will allow to…”, “…about 5–20% of RA patients are resistant to all proposed treatments have been defined as “difficult to treat RA”

We modified the text

  1. Please correct “Authors” to “authors” in multiple sentences. 

We modified the text

  1. “to prescribe the best drug for the right patient early in the disease process” is not one word.

We changed the text
